# The κ-Model under the Test of the SPARC Database

**Gianni Pascoli**

Faculté des sciences, Département de physique, Université de Picardie Jules Verne (UPJV), 33 Rue Saint Leu, 80000 Amiens, France; pascoli@u-picardie.fr

**Abstract:** Our main goal here is to conduct a comparative analysis between the well-known MOND theory and a more recent model called the κ-model. An additional connection, between the κ-model and two other novel MOND-type theories, Newtonian Fractional-Dimension Gravity (NFDG) and Refracted Gravity (RG), is likewise presented. All these models are built to overtake the DM paradigm, or at least to strongly reduce the dark matter content. Whereas they rely on different formalisms, however, all four seem to suggest that the universal parameter, $a_0$, appearing in MOND theory could intrinsically be correlated to either the sole baryonic mean mass density (RG and κ-model) and/or to the dimension of the object under consideration (NFDG and κ-model). We then confer to parameter $a_0$ a more flexible status of multiscale parameter, as required to explain the dynamics together in galaxies and in galaxy clusters. Eventually, the conformal gravity theory (CFT) also seems to have some remote link with the κ-model, even though the first one is an extension of general relativity, and the second one is Newtonian in essence. The κ-model has been tested on a small sample of spiral galaxies and in galaxy clusters. Now, we test this model on a large sample of galaxies issued from the SPARC database.

**Keywords:** SPARC database; galaxy; MOND; Newtonian Fractional-Dimension Gravity; refracted gravity; κ-model; dark matter

## 1. Introduction

As is well known, all the studies conducted on galaxies and galaxy clusters lead to the seemingly firm conclusion that a significant portion of the mass in the Universe seems to be hidden from the view of the observers. This invisible (non-baryonic) matter is called dark matter (DM). It is true that DM is a simple and economic hypothesis. However, the major problem with this paradigm is that the dark matter/baryonic (DM/B) mass ratio is incredibly huge, of the order of six. It is not simply an addition of a small quantity of missing matter to a dominant form of visible (baryonic) matter. In fact, it is the opposite, and the baryonic component eventually appears negligible in the Universe. This situation seems to be surprising and even rather uncomfortable, the visible sector being explained by an undefined invisible sector about which we know nothing. De facto, the explanation of the flatness of the rotation spiral galaxy curves with DM is fully indirect. A very pertinent parallel can be drawn with the phlogistic theory, a dominant theory in nascent chemistry during the 18th century. The phlogistic hypothesis was based on the existence of an illusive "substance" (the phlogiston) with indeterminate properties and thus without real physical support. The theory of phlogiston was finally disproved by the French chemist Antoine-Laurent Lavoisier through a series of experiments in the late eighteenth century. Is it the fate that awaits DM? At present, the existence of DM is inferred only through gravitational effects. A direct proof is missing from both an observational and an experimental point of view.

The MACHOs (Massive Compact Halo Objects), possibly detected through gravitational microlensing in the Galactic halo, have been ruled out as a dark matter candidate [1]. Another interesting interrogation is that if DM particles really exist, these particles can very possibly decay. Strangely enough, X-ray space telescopes like Chandra, XMM-Newton,

and Fermi have not observed any excess of DM decay [2]. Eventually, a major issue for DM is the tantamount difficulties of observing DM particles in laboratory. Large classes of candidates have been suggested following highly speculative theoretical models, such as Hidden-Sector Dark Matter particles, completely neutral under Standard Model forces, but interacting through a new force; or still Ultra-Light Dark Matter particles with predicted masses from 10–22 eV to about a keV that can be produced during inflation or phase transitions in the very early Universe [3]. However, the existing dark matter experimental programs are now more reasonably focused on weakly interacting massive particles (WIMPs) [4–6]. Unfortunately, the conclusions of all these very costly studies are always negative. All direct detections have come up empty. The persisting non-detection in space and in the laboratory of DM in spite of very intense efforts is rather discouraging. A simple but very frustrating conclusion is that if DM interacts uniquely gravitationally with baryonic matter and definitely not through one of the other known three forces (the strong, weak, or electromagnetic forces), we might never detect it. Another possibility is that DM interacts with itself and with baryonic matter but via an unknown (fifth) force. In spite of all that, DM remains the leading explanation for the dynamics of galaxies, very likely for its high flexibility adaptable to various situations (galaxies, galaxy clusters, and cosmology). This view can unfortunately persist for a very long time because the DM paradigm seems to be unfalsifiable. Yet, a good question posed by McGaugh [7] is the following: Is it a missing mass problem, or rather an acceleration–velocity discrepancy when observing the galaxies? Indeed, the mass is indirect data, in contrast to acceleration which can be directly measured. Di Paolo and coauthors [8] have remarked that there exists a mysterious link between DM and the baryonic component. In fact, this link is easily explained if DM is a property of the baryonic mass itself.

Alternatively, a lot of authors have tried to circumvent the trouble by exploring paths other than DM. Without DM, it is true that the Newtonian theory of gravity, and even its basic relativistic version, i.e., the general relativity, seem to fail on galactic scales. The first model that was developed in this sense is the Modified Newtonian Dynamics or MOND [9–11]. Remarkably, the basic idea of this model is thus as simple and economic as DM concerning the theoretical background. The initial aim was to explain the flatness of the rotation velocity curves of the spiral galaxies uniquely with the help of the observed baryonic matter. In MOND, the second law of Newton ($ma = F$) is modified in the very low regime of acceleration, $a \leq a_0$, $a_0 \sim 1.2 \times 10^{-10}$ m s$^{-2}$, being a universal constant. MOND replaces acceleration $a$ by $\frac{a^2}{a_0}$. Assuming then a test particle surrounding attractive mass $M$, with a circular orbit of radius $r$ and with $F = \frac{GM}{r^2}$, we have $\frac{a^2}{a_0} = \frac{GM}{r^2}$ or $a = \frac{\sqrt{GMa_0}}{r}$. For velocity, we directly obtain $v^2 = ar = \sqrt{GMa_0} = Const$. This leads to the flatness of the observed rotation curves of spiral galaxies but, more importantly, results in the Tully–Fisher law in a very natural manner [12].

Furthermore, MOND is sustained by the empirical Renzo's rule. The empirical Renzo's rule highlights the correspondence between detailed features observed in the observational rotational curves of spiral galaxies and the same features seen in their Newtonian counterparts [13]. This statement, that the observational rotation profiles seem to be a magnification of the Newtonian counterparts, appears quite natural when baryonic matter dominates the mass, but not if DM is the dominant form of matter. Another strong support for MOND, as seen above, is direct deduction, within a calculation that takes just a few lines, of the Tully–Fisher relation. These two facts are difficult to explain within the DM paradigm, except in an ad hoc manner. Eventually, MOND predicted, well in advance, the profile of the rotation curves in the case of low surface brightness galaxies (LSB), once again a feat not possible for DM [13]. However, the MOND phenomenology fails to explain the dynamics of galaxy clusters. A natural remedy was found by adopting a multiscale approach [14][1]. In any way, as for gravitational lensing and cosmology, the classic modified-(gravity+inertia) MOND in its initial form [9] is not applicable. Various relativistic versions of MOND (RMOND) have been proposed, making clear predictions regarding gravitational lensing and cosmology. The latest in date is that of Skordis and Złośnik [15]. The latter

version of RMOND reproduces the galactic and lensing phenomenology and also the key cosmological observables[2].

Another well-known modified gravity theory is the covariant scalar–vector–tensor modified gravity (MOG) built by J. Moffat [17]. MOG is based on a $D = 4$ pseudo-Riemannian metric, a spin-1 vector field, a corresponding second-rank skew field $B_{\mu\nu}$, and eventually three dynamical scalar fields, $G$ (the gravitational constant), $\omega$ and $\mu$. The heavy price to be paid is the addition of extra vector and scalar fields to the gravity field. On the other hand, in MOG, gravitational constant $G$ is assumed to vary with space and time. Moreover, the introduction of new fields means that new particles are surreptitiously hypothesized. We are not far from DM with its elusive particles, even though MOG is much more subtle than DM because the particles in MOG are virtual and may not be directly observable in the laboratory. MOG has been largely applied with some success to spiral galaxy curves, to galaxy clusters, to gravitational lensing and eventually to cosmology [18]. RMOND and MOG are the two main models built to get rid of DM fairly efficiently. These two models are the only ones that have been extensively studied and involved in concrete comparisons with the observational data. Unfortunately, with the relativistic extension of MOND, or with its main concurrent MOG, one moves away from the beautiful simplicity of the Newtonian mechanics and even of general relativity. Let us note that RMOND and MOG appear very much alike. Thus, the major pitfall of RMOND and MOG is the introduction of other subsidiary extra scalar and vector fields that have not been tested in laboratory.

A broad number of other models also exist, but they have been more sporadically applied to real situations. Conformal gravity theories (CFTs), which are compelling alternatives to general relativity theory, have been claimed to explain the observed flat rotation profiles of spiral galaxies, without invoking DM or other exotic modifications of gravity [19,20]. Nevertheless, the extension of this type of models to the field of cosmology appears to be questionable. Thus, it seems that Weyl CFT[3] cannot accurately describe the stated lensing observations without again considering dark matter [21]. Eventually, another very different way is to conceive gravity not as a conventional interaction, but rather as an emergent property [22]. In this case, gravity is seen as an entropic force, i.e., closely related to thermodynamics. The testing of this hypothesis in the galaxy world is underway.

Are there other options to get rid of DM? We can answer this question in the affirmative. Very recently and quasi-simultaneously, a lot of new models have been proposed in the following original, even though speculative, ways [23–27]. These models sound similar, even though they use different formalisms (see Table 1). The aim is then to satisfy a principle of parsimony in the introduced concepts. It is indeed about three different strategies which, however, share a number of common features. All these ideas are new and still need deep understanding.

One very aesthetically effective strategy is to assume that spacetime is multifractal in nature. This property is revealed in the most prominent quantum gravity theories in a natural manner [28]. This concept of fractional- imension space applied to Newtonian gravity has been suggested as an alternative to DM [23,29–32]. In the latter work, a connection is established between the Newtonian Fractional-Dimension Gravity (NFDG) with MOND. The MOND acceleration constant $a_0$ can be related to natural scale length $l_0$ in NFDG, i.e., $a_0 \sim \frac{GM}{l_0^2}$, for any astrophysical structure of mass $M$, and the deep-MOND regime appears in regions of space where the dimension is reduced to $D \sim 2$.

The second strategy is Refracted Gravity [33,34]. Refracted Gravity mimics dark matter by introducing a gravitational equivalent to a permittivity, seen as a monotonic function of the local mean volumetric mass density. This function is parametrized by three coefficients which are free as in the case of DM, but which are expected to be universal, in contrast to DM where the parameters are free and, additionally, different for each galaxy. Once again, even if this second strategy apparently relies on a very different formalism than NFDG, both share strong links with MOND.

We turn now to the third strategy, i.e., the $\kappa$-model. The aim of the $\kappa$-model is to reflect on the ways the mean volumetric mass density (estimated at a very large scale) surrounding a given observer can modify his view of the Universe.

In the framework of this model [16,25–27], it is hypothesized that it is the perception of the observer modified by his environment (the local mean volumetric mass density, calculated at a very large scale around him) that creates the observed anomalies and also their proper experience of gravity (with the current need to call for a hypothetical dark matter in order to explain these anomalies). This idea is speculative, but it strongly resembles the models for which we provided an overview above [23,24]. However, one point of difference is that the effects described in the $\kappa$-model are only apparent depending on the observer (excepting the spectroscopic velocities whose measurement is universal; see [16] par. 2 eq. 10). It is almost as if we are looking at any object through a perfect, even though fictive, optical device (such as an aberration-free flat superlens[4], but without being aware of the presence of this device (which obviously does not exist)). Clearly, the object does not change but both its apparent size and velocity can now appear magnified from the point of view of a distant observer. Admittedly, both the inertia and the gravity seem to be modified in the $\kappa$-model, but it is a pseudo-modified gravity, it is not of the same nature as real modified gravity as introduced, for instance, in MOG or RMOND. Moreover, in the $\kappa$-model, the gravitational constant (and the speed of light) locally measured by any observer is invariant. The gravitational constant, the speed of light, and all physical constants are universal in the $\kappa$-model. To make variable a constant in physics, in our case, $G$ could require other constants to be variable (for instance, the speed of light) with possibly unpredictable consequences. Furthermore, no new field or exotic particles undetected in the laboratory are assumed in the $\kappa$-model. We think that it is a very important point that obeys a principle of parsimony. Eventually, even though the $\kappa$-model is Newtonian in essence, its great advantage is that it can be naturally made relativistic. A first draft of what might be a relativistic version of the $\kappa$-model is presented in reference [25]. However, in a galaxy, velocities $v$ of stars and gas are low compared to speed of light $c$ (ratio $\frac{v}{c} \sim 10^{-3}$), and the nonrelativistic approximation is sufficient, especially on the outskirts of galaxies where gravity is weak. The same arguments also apply to MOND. For MOND, a notable relativistic version has, however, been proposed [15]. Nevertheless, the elegant simplicity of the initial version of MOND has unfortunately disappeared in the operation. At Newtonian level, the $\kappa$-effect is mimicked by an apparent local scaling transformation applied in an Euclidean space [16,26]. In a Riemannian structure of space, a local scaling transformation could be applied exactly in the same manner. Eventually, let us note that the multiscale approach already suggested in [14] is directly included in the $\kappa$-model, which assumes that the larger the characteristic dimension (the scale) of a system, the weaker the local mean volumetric mass density and the stronger the magnification [16].

In order to avoid any misunderstanding, three velocities are defined in the $\kappa$-model: the Newtonian velocities, $v_{New}$, which are directly calculable from the mean surface mass density profiles, but which are virtual and not measurable, the radial velocities, which are given by $v_{rad} \sim \kappa^{\frac{1}{2}} v_{New}$ (observationally universal spectroscopic velocities, $v_{spec}$), and the tangential velocities which are given by $v_{tan} \sim \kappa \, v_{New}$ (observationally the proper motions). Following a more mathematical approach within the formalism of bundles, the Newtonian velocities are "located" in the base (not reachable) and both the measurable radial and tangent velocities are "located" in a sheet, attached to a given observer in the bundle situated "above" the base [16]. The latter mathematical considerations are shortly developed in an upcoming paper. We are only concerned here with the observational aspect.

**Table 1.** The synoptic table below summarizes the applicability domains of the different models discussed in this paper.

| Model | Main Features |
|---|---|
| MOND | Very low acceleration, $a \lesssim a_0 \sim 10^{-10}$ m s$^{-2}$ |
| $\kappa$-model | Very low mean mass density $\lesssim 0.15\ M_\odot pc^{-3}$<br>Geometry of the matter distribution (bulge, disk)<br>Compactness (stars, gas) |
| NFDG | Variable dimension of the matter distribution, between $D = 3$ (sphere) and $D = 2$ (disk) |
| RG | Very low mean mass density $\ll 0.17\ M_\odot\ pc^{-3}$; geometry of the matter distribution (bulge, disk) |

## 2. Calculation Details

In the SPARC catalog [35], each galaxy is usually identified by three independent main components for the densities: the bulge labeled $b$ in the following, the stellar disk labeled $d.st$, and the gaseous disk labeled $d.g$. This hierarchy is also preserved in the $\kappa$-model where both the geometry and the relative values taken by the mean densities (compact masses for the stellar component or diffuse masses for the gaseous component) are now playing a new role by their involvement in a magnification factor at a very large scale. A similar idea appears in the NFDG theory, but it is the dimension of the matter distribution that plays a major role. Let us note that the so-called $\kappa$-effect (a retranscription of the DM-like effect), said in a practical way, is a "huge-volume-effect" and it only occurs at a very large scale; it is inexistent at the solar system level (a bit like the quantum effects are fully imperceptible at the macroscopic level). In the framework of the $\kappa$-model, the relationship associating the corresponding (fictive) Newtonian velocities to the measured spectroscopic velocity is [16,26,27]

$$v_{spec} = \left( \frac{\kappa_{M_t}}{\kappa} \right)^{\frac{1}{2}} \left[ \frac{\kappa_{ref}}{\kappa_{Mst.b}} v_{b.st}^2 + \frac{\kappa_{ref}}{\kappa_{M_{d.st}}} v_{d.st}^2 + \frac{\kappa_{ref}}{\kappa_{M_{d.g}}} v_{d.g}^2 \right]^{\frac{1}{2}} \tag{1}$$

where each peculiar velocity is weighted by a $\kappa$-ratio. The origin of the $\kappa$-ratios results from the need to take into account explicitly both the matter distribution dimension (bulge or disk) and the compactness of this matter (stars or gas). In the $\kappa$-model, all these coefficients are directly linked to mean volumetric mass densities $\rho$ by a simple and universal relationship (ln denotes the natural logarithm)

$$\frac{\kappa_1}{\kappa_2} = 1 + ln \left[ \frac{\rho_1}{\rho_2} \right] \tag{2}$$

with the necessary condition of $\frac{\rho_1}{\rho_2} > 1$. The indexes "1, 2" run on all the mentioned indexes. Relation (2) is called universal in the sense that this relation is valid regardless of the type of galaxies, and also for galaxy clusters [16,26]. In MOND, the analog of $\kappa$ is not a logarithmic function of the density but a rational function of the distance [9–11] (but both are sensibly equivalent in the case of an exponential distribution of matter). In Relation (1), indexes $ref$, $M_t$, $M_{b.st}$ and $M_{d.g}$, respectively, designate the reference value for the density, maximum value $M$ of the total density, $t$, (stellar bulge, $b.st$, + stellar disk, $d.st$, + gaseous disk, $d.g$) estimated at the center of the galaxy, and maximum value $M$ of each of the independent components, also estimated at the center of the galaxy. Non-indexed coefficient $\kappa$ is the local one (there resides the observer who feels the gravitational field). For practical purposes concerning the disk components, density $\rho$ can be expressed as a function of the observable surface mass density (indirectly obtained from the brightness measurement), i.e., $\rho = \frac{\Sigma}{\delta}$ with thickness $\delta$, the latter quantity here assumed to be constant throughout a galaxy disk. Apparently, the thickness of the disks seems to play a role in the $\kappa$-model,

very similarly to what is assumed in the *NFDG* model, even though in the *NFDG* model it is the dimension of the mass distribution that intervenes instead of the thickness [23,30]. Variable thickness along a galactic radius in the $\kappa$-model could have a close connection with variable dimension $D$ in the *NFDG* model. However, it is not as simple as it appears, and we return to this issue later. The magnification coefficients of the active mass comprising both the stellar and gaseous components are expressed separately, respectively, $\frac{\kappa_{ref}}{\kappa_{M_{b.st}}}$, $\frac{\kappa_{ref}}{\kappa_{M_{d.st}}}$ and $\frac{\kappa_{ref}}{\kappa_{M_{d.g}}}$, but are still calculated with the same universal relationship (2). When the mean surface mass density is larger than $500\ M_\odot\ pc^{-2}$, a saturation effect appears for $\frac{\kappa_{ref}}{\kappa_M}$, and then, in all the cases, we put this factor invariably equal to 0.45, as provided by Relation (2). However, in a few rare situations, especially for galaxies with a big bulge, and in order to adequately fit the observational profiles in the inner regions, we adjust factor $\frac{\kappa_{ref}}{\kappa_{M_{b.st}}}$ to a value between 0.45 and 1. An explanation to this statement is that, in fact, Relation (2) is valid for a thin disk, but not for a 3D bulge. At this level, a clear reference to the NFDG model where the dimension plays a major role can be noticed. Another explanation is that the bulge of a spiral galaxy is a very complex system where the stellar orbits are randomly oriented. Then, we know that a severe velocity dispersion, larger than $\sim 50\ km/s$ a few kpc from the center, can strongly affect the extraction of pure rotation velocity (see, for instance, reference [36] for the Milky Way). The part of the cylindrical rotational support in the inner regions of a spiral galaxy is generally difficult to estimate when the bulge is dominant.

The following is a fundamental question: How many free parameters are used in the $\kappa$-model? We know that in physics, the fewer the number of parameters, the better the model. Yet, by consulting Relation (1), we see immediately that four parameters (the $\kappa$-ratios) appear. Following the parsimony principle, it is not a "good" model. In fact, once the density in the bulge in the stellar and gaseous disks is provided, the $\kappa$-ratios, which are directly issued from observational data, are automatically determined, there are no longer free parameters and the $\kappa$-model eventually becomes parameter-free (the only parameters usually being the observables, i.e., the surface brightness, the inclination, and the distance, even though, unfortunately, not very well known in some cases). This is in strong contrast with DM where two or three free external and arbitrarily chosen parameters are introduced to just obtain the expected results. However, given that the $\kappa$ ratios are dependent on the densities, the parameters in the $\kappa$-model can now vary from one object to another, and this confers some flexibility to the model with no violation of the parsimony principle. For instance, the $\kappa$-model has been applied with success to the physics of galaxy clusters [16]. The mean mass density in a galaxy cluster is lower by three orders of magnitude compared to the mean mass density in a galaxy. The $\kappa$-model is then naturally a multiscale model (or density-dependent scale model), like the one proposed in [14] for the application of MOND to galaxy clusters. The difference is that in the $\kappa$-model, the scaling is not imposed, but appears in essence, taking its origin in the hierarchy of the mean mass densities. By contrast, MOND [9–11] with just one universal parameter or even Refracted Gravity [33,34] with three universal parameters seems to be too rigid. On the other hand, the $\kappa$-model can naturally be made relativistic [25], making possible its extension to cosmology, especially to the analysis of the fluctuation density in the CMB. In this case, it is the density anisotropies-to-mean density ratio which intervene in Relationship (2). The latter very important topic will be examined in a next paper.

Now, if we want to compare the $\kappa$-model and MOND, we must define a reference point for mean mass density $\rho_{ref}$. Unfortunately, this quantity is only indirectly determined by ratio $\frac{\Sigma}{\delta}$ (surface mass density, $\Sigma$, over the disk thickness, $\delta$, in a spiral galaxy such as the Milky Way). The link between acceleration parameter $a_0 \sim 1.2 \times 10^{-10}\ m\ s^{-2}$ of MOND and reference surface density $\Sigma_{ref}$ is

$$\Sigma_{ref} = \frac{a_0}{2\pi G} = 152\ M_\odot\ pc^{-2} \tag{3}$$

Let us note that this value is relatively close to the galactic surface mass density estimated in the solar region ($\sim 70\ M_\odot\ pc^{-2}$) (in comparison with the high range of surface densities seen in a disk galaxy, varying from $\sim 1000\ M_\odot\ pc^{-2}$ in the inner regions, 1 $kpc$ from the center, to $\sim 1\ M_\odot\ pc^{-2}$ in the outskirts, 20 $kpc$ from the centre). Taking into account the fact that the range of mean mass densities is very extended in the Universe, this appears indeed very odd if we see parameter $a_0$ as a cosmological parameter; because in this case, we must assume that our situation in the Universe is privileged. In reality, the $\kappa$-model easily explains this rather strange coincidence. We chose this reference taking into account our position in the galaxy, but it is not particularly remarkable. Another observer, located elsewhere, takes their own reference. In Relation (1), offering a universal result in the framework of the $\kappa$-model, their measurements would lead exactly to the same results for the spectroscopic velocities as ours, even despite their proper local reference for the mean mass density.

Figure 1 displays a panel of velocity profiles for MOND and the $\kappa$-model in the schematic situation of a disk of matter where the mean surface mass density varies exponentially (the thickness is assumed to be constant following radius $r$). The comparison between MOND and the $\kappa$-model shows that logarithmic Relation (2) is a very good choice. In MOND, function $\mu(r)$ plays a very similar role (see Equations (7) and (8) of reference [13]), even though in MOND $\mu(r)$ is not a logarithmic function but a simple rational function. We can note that the $\kappa$ effect (or the MOND effect) plays a decreasing role when progressing from low mass surface density (LSB galaxies) toward high mass surface density (HSB galaxies), as confirmed by the observations. This finding, naturally explained with MOND or the $\kappa$-model, remains unexplained in DM. A difference between MOND and the $\kappa$-model is, however, perceptible for the schematic representation of a so-called super spiral [37]. For high surface density ($\Sigma_M \sim 10,000\ M_\odot\ pc^{-2}$, see, for instance, reference [38]), the $\kappa$-model curve is located more than 200 km/s above the MOND profile for terminal velocity (Figure 1d).

In the more concrete cases, the situation is obviously different from the previous trial examples with a simple exponential disk. In reality, we encounter in the SPARC catalog a number of situations where it is not possible to fit the mean surface mass density of both thin stellar $\Sigma_{d.st}$ and gaseous $\Sigma_{d.g}$ disks by just adopting a simple exponential fit. In these situations, we have to add to the exponential component one or sometimes two decentered Gaussian components. Velocity curve $v_d(r)$ is then deduced from the self-evident, more general formula (still assuming an axisymmetric disk) valid for one component (stars or gas):

$$\frac{v_d^2(r)}{r} = G \int_{\Omega_\infty} dx\,dy\, \frac{\Sigma\left(\sqrt{x^2 + y^2}\right)(r - x)}{\kappa(r)\left[(r - x)^2 + y^2\right]^{\frac{3}{2}}} \tag{4}$$

As the first step, the operational method consists in fitting the Newtonian velocities available from the SPARC catalog for the distributions of stars and gas, taken individually, and for each galaxy. In simple terms, we fit the dashed red (stars) and dashed green (gas) curves of Appendix A (Figures A1–A10). In this case, Relationship (4) is applied with $\kappa(r) \equiv 1$ (this is the usual Newtonian level). Secondly, the same Relationship (4) is again used, but incorporating this time coefficient $\kappa(r)$ that depends on the volumetric mass density (Equation (2)). This second step automatically provides the corresponding $\kappa$-model curves. The great benefit of the method is that all parameters are internal to the theory and supported by the sole observational data, essentially the baryonic mass density. There is no arbitrary parameter such as the ad hoc DM/B ratio in DM.

Eventually, when a bulge is present, a de Vaucouleurs formula [39] is used to fit the surface mass density of the bulge. Two other parameters intervening in the $\kappa$-model are still the thickness (scale height) of the stellar (thick) disk, $\delta_{st}$, and that of the gaseous (thin) disk, $\delta_g$. For all the galaxies under study (SPARC catalog), these parameters are taken to be equal to the reference values estimated for the Milky Way in the vicinity of the Sun,

respectively, $\delta_{\odot.st}$ and $\delta_{\odot.g} \sim 0.5\,\delta_{\odot.st}$. Given that the galaxies are diversely oriented with any inclination angle, these parameters are difficult to estimate and certainly variable along a galactic radius. Our analysis of the SPARC galaxies seems to indicate a neat trend where the thickness decreases when moving from the core regions to the outskirts in flattened galaxies.

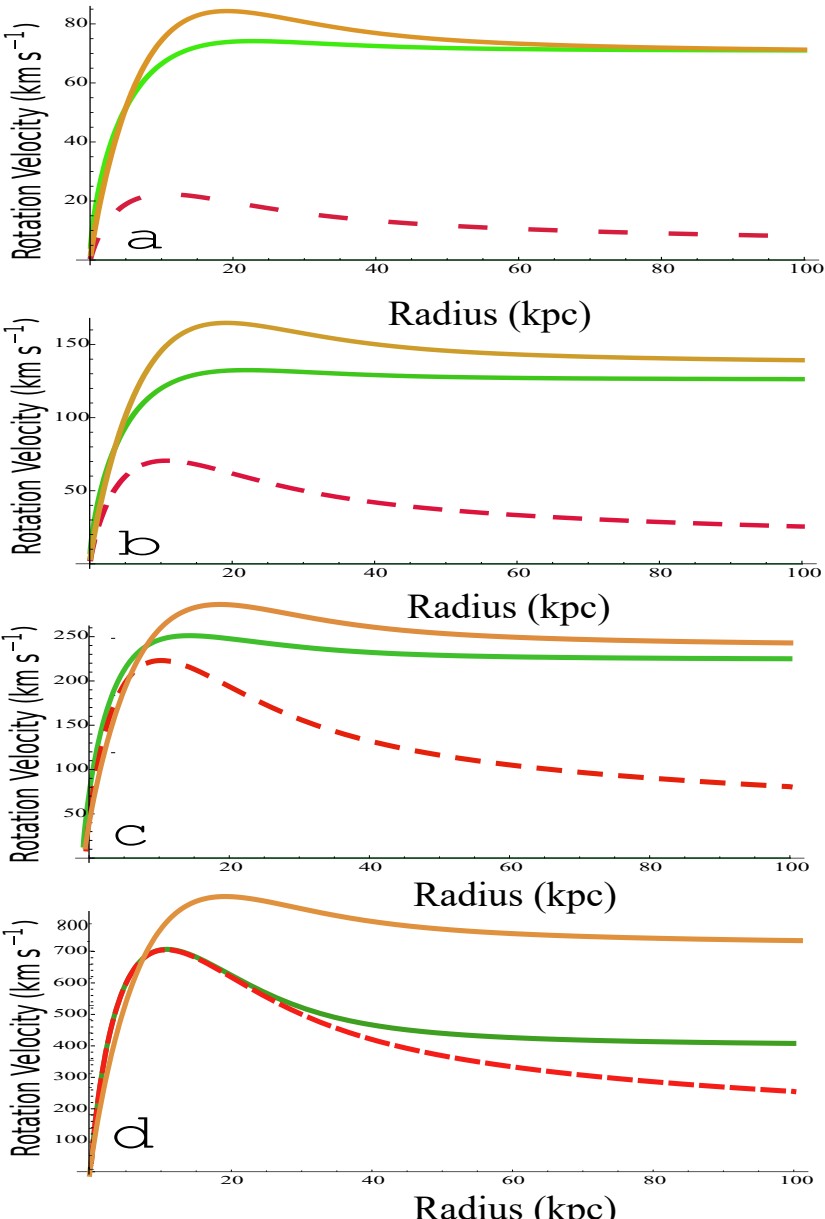

**Figure 1.** Schematic galaxy velocity curves fitted with a simple exponential surface mass density profile of uniform thickness (in the approximation of the thin disk). MOND is the green line, and the $\kappa$-model is the amber line. The dashed red line is the Newtonian curve (baryons). The reference values for the surface density are $\Sigma_{ref} = \frac{a_0}{2\pi G} = 152\ M_\odot\ pc^{-2}$. (**a**) $\Sigma_M = 10$, (**b**) $\Sigma_M = 100$, (**c**) $\Sigma_M = 1000$,(**d**) $\Sigma_M = 10,000\ M_\odot\ pc^{-2}$.

Globally, for a mean orientation of $45°$, the thickness along the line of view is increased by a factor of $\sqrt{2}$. In this case, the rotation profiles provided by the model have to be magnified by a few percent. In fact, the logarithmic function flattens the density ratios in Relation (2) and the influence of the variation of the thickness has a strongly reduced, even though not negligible, impact on the corresponding $\kappa$ ratios (of the order of 10% for a thickness variation of a factor of 2. For orientations larger than $45°$, the magnification can

obviously be much larger than 10%). Let us note that in other models where the density is assumed to play a role, for instance, in [24,30], the conclusions should be very similar when applied to a large sample of galaxies such as the SPARC database. The thickness of various types of spiral galaxies has been estimated by different methods [40–42]. For irregular dwarf galaxies, the situation appears relatively confusing, but the latter category can exhibit quasi-round galaxies with high mean thickness [40]. The measurement of the thickness seems to provide values of the order of $\delta_{\odot.st}$ or $\delta_{\odot.g}$ to within a multiplicative in the range from 0.25 (in the outer regions) to 4 (in the inner regions toward the bulge, if existing) compared to the reference values, independent of the size of the galaxy (with a few exceptions for the very small galaxies, where smaller values for the thickness are favored). Thus, a positive point is that an estimate of the thickness can be reached in the framework of the $\kappa$-model. However, in Appendix A (Figures A1–A10), for all the galaxies and for the sake of homogeneity, the thickness is taken as invariable throughout the stellar and gaseous disks. The corresponding values are indicated in each individual figure. Taking into account a variable thickness would make it possible to obtain better profiles, which is a topic of work that remains to be conducted.

Additionally, let us specify that the observations rather provide non-monotonous galactic rotation profiles. Nevertheless, it is illusory to try to perfectly fit the rotational curves with their delicate patterns of bumps and wiggles. Very likely, these patterns are caused by the presence of spiral arms or portions of rings, a variable thickness or inclination, not taken into account by assuming smoothed axisymmetric and monotonous density profiles. Even DM with two or even three external parameters cannot make that[5]. One of the better DM methods, built on the Einasto profiles with three ad hoc parameters in the fits, is discussed in reference [45]. We can see that the fine details cannot be adequately fitted (see, for instance, NGC6015, NGC 7793, NGC3726, IC4202, NGC0289, UGC06787, etc.). In any case, a lot of physical parameters are very poorly known: the inclination of the galaxy (moreover, very likely variable along the galactic radius), the mass-to-light ratio, the thickness along the line of sight, the distance, etc. We must add that the observational profiles can substantially differ in some cases from one author to another, sometimes by more than 20 km/s. We can compare two different catalogs, for instance, that of Sofue [46] with SPARC [35], when the rotation curve for the same galaxy is presented (see especially NGC 2903 where a discrepancy of 40 km/s can be noted). Even for the Milky Way, in the vicinity of the Sun, divergences also exist [47]. Let us note that the DM paradigm could, however, be produced in agreement with any inclination by adequately adjusting the DM/B rate. In contrast, both MOND and the $\kappa$-model apparently fail if the inclination is not accurately estimated [26]. An example where the inclination factor can sometimes play an important role in the determination of the rotation velocity profiles is given in [48]. In the paper, it is shown that the inclination can vary by $20°$ according to the authors eventually favoring this model rather than another one. Eventually, we can say that, unfortunately, the determination of the inclination is not the sole issue. Additionally the gas and the stars in a galaxy, following their types, do not rotate in the same manner, the velocities are not circular, the galaxy disks are not symmetric, etc. The multiple consequences on the observational profiles are difficult to estimate. This is why various observational techniques can lead to different profiles for the same galaxy.

In spite of these difficulties, and in order to make a valuable comparative analysis between different theoretical models, the idea is to use the same set of extended data. For instance, the SPARC catalog seems, in this case, necessary. This catalog gathers a large and homogeneous sample of rotation profiles. A very good point of the SPARC database is that it represents a uniform estimate of the surface densities of galaxies, starting from Spitzer near-infrared data [35]. Then, our procedure consists in starting from the mean fits of the Newtonian curves, and then the mean fits for the observed rotational curves can be deduced. In some cases, the DM fits seem to be much more impressive [44,45], but a major drawback for a physical model is that the DM technique of fitting is not at all predictive. Then, starting from any Newtonian curve (even false), we can build any "good" predicted

profiles, obviously by adding the "good" rate of DM. Admittedly, MOND and $\kappa$-model profiles are generally of lesser quality, but in most cases, both of them produce a good trend for the fits compared to the observational rotation curves. Let us specify again that the latter ones, impaired by various biases, are not perfect, either.

## 3. Results

### 3.1. MOND versus $\kappa$-Model

The results of calculations for the individual galaxies in the SPARC catalog are collected in Appendix A. The galaxies are classified in alphabetical order to facilitate the research. For the disks, it is assumed that the thickness is constant along the radius of the galaxy. In most cases, the thickness is taken to be equal to the corresponding reference values taken at the Sun position in the Milky Way, for both the disks, stellar, $\delta_{\odot.st}$, and gaseous, $\delta_{\odot.g}$. In view of the results, the first general remark is that the $\kappa$-model is clearly as predictive as MOND[6]. For both models, the results statistically deviate by less than 10% as for the prediction of terminal velocities (Figure 2).

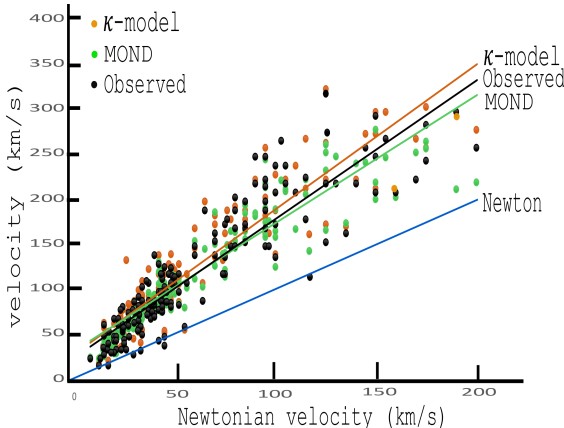

**Figure 2.** The terminal velocities for the sample of galaxies (SPARC database). MOND is in green, the $\kappa$-model is in amber, and the observed velocities, provided by the SPARC database, are represented in black. Linear regression lines are also represented.

Examining the individual cases by browsing Appendix A, we can see that the $\kappa$-model leads to predictions similar to the MOND phenomenology, even though in some cases, the profiles are not quite identical. Moreover, a comparison with the observational profiles shows that the predicted curves for both MOND and the $\kappa$-model do not perfectly match the observations[7]. Referring to Appendix A, we see that the theoretical curves predicted by MOND or the $\kappa$-model can be indifferently located slightly above or below the observational curves. However, there are remedies for this. First, in the outer regions, the predicted curve is quite often located above the observational one. In MOND, the bias can then be corrected by EFE (External Field Effect) [52]. Likewise, this bias could be corrected by a diminushing thickness of the disks (at constant surface density $\Sigma$) in the $\kappa$-model. In contrast, there exist a number of cases where the predicted curve is located below the observational one, especially in the innermost regions of galaxies (for instance, in the more striking cases: F563-V2, F568-1, F568-V1, F571-8, F579-V1, F583-1, NGC 2915, NGC 3992, NGC 5907, NGC 5985, NGC 6674, UGC 00128, UGC 00731, UGC 02259, UGC 6446, UGC 06667, UGC 07399, UGC 7490, and UGC 8286). It is interesting to note that the magnitude of this bias is very similar in both MOND and in the $\kappa$-model for a given galaxy. It is very possible that a modification of the inclination in MOND (and also in the $\kappa$-model) could partially remove the discrepancy in the inner regions for these galaxies. As demonstrated in [48], the modification of the inclination can substantially modify the profile of the rotation curve. However, this effect appears rather systematic throughout Appendix A in the inner regions. In other words, the measurement of the inclination is systematically biased in the inner of spiral galaxies and, strangely enough, always in the same direction. This hypothesis

is hardly acceptable. The fact that the MOND curve is located below the observational one results from the fact that acceleration $a$ is equal to or larger than critical value $a_0$ in the inner regions. In this case, we are in a domain where the Newtonian regime is still supposed to be valid. To save MOND, we can then assume that parameter $a_0$ is larger in the inner region, but then this parameter is no longer universal. Another solution is that the baryonic mass-to-light ratios are largely underestimated (by a factor of two) in the inner regions of the quoted galaxies. Eventually, a more credible explanation is to imagine that some non-exotic form of DM exists in the innermost regions of galaxies. We could invoke, for instance, a neutrino species with mass $\sim$eV (but in acceptable quantity with $DM/B \sim 2$). A very similar idea has been expressed regarding the inner regions of galaxy clusters [53]. This hypothesis appears admittedly reasonable; however, the $\kappa$-model can result in another natural solution. In the $\kappa$-model, the leading role is not held by a fixed parameter, i.e., acceleration $a_0$, but by the mean volumetric mass density. This hypothesis makes the $\kappa$-model more flexible than MOND. Appendix A displays the results under the reductive hypothesis of a constant thickness throughout the galactic disks. However, an increase in disk thickness (at constant mean mass surface density) in the inner regions of spiral galaxies could help to lessen the discrepancy. An interesting conclusion is that the $\kappa$-model could hence help to obtain an estimation of the mean thickness, a parameter difficult to derive from the observations. In any case, in the cases mentioned above, even if the $\kappa$-model produces imperfect fits in the innermost regions of these galaxies, we can see that the terminal velocities are correctly predicted. A simple response to these statements is that empirical Relationship (2) is very well adapted to a thin disk, but far less applicable to a thick disk or a 3D bulge.   For the galaxies listed above, where a discrepancy exists between the $\kappa$-model (or equivalently MOND) and the observational data, a comparison with DM profiles with two or three (ad hoc) parameters (as in reference [44]) appears very interesting. Examining the cases displayed in Figure 6.10 of [44], we can see that the underlined discrepancy also persists in some of the cases, even though slightly lessened (see, for instance, the rotation profiles for F568-1, F579-V1, NGC 2915, NGC 5907 and NGC 6674). F571-8 is a pathological example where MOND, the $\kappa$-model, and DM provide very similar profiles, paradoxically enough far from the observational one in the outer regions. The three theoretical profiles, even though very similar, are located 50 km/s below the observational profile in the outer regions. In any way, we know that trying to predict the rotation velocity curves with better statistical precision than 10% (and even, in some pathological cases, the incertitude can rise to 20%) appears unwarranted, considering the dispersion in the observational data coming from various sources. That matter aside, in the framework of the $\kappa$-model, at least we have a fairly good estimation of terminal velocities (Figure 2 and see also Appendix A for the individual cases), a conclusion that cannot be reached using the ad hoc DM methodology. Obviously, the flexibility of the $\kappa$-model caused by taking into account variable thickness in the stellar and gaseous disks would allow for fixing the residual discrepancy between the theoretical curves and the observational ones. In the same vein, this statement is rather attractive because it implies that the $\kappa$-model could predict the variation of the thickness in spiral galaxies along a radius. These data are indeed difficult to obtain by sole observation.

### 3.2. Newtonian Fractional-Dimension Gravity

Newtonian Fractional-Dimension Gravity (NFDG) is an extension of the laws of Newtonian gravitation to lower-dimensional spaces, including those with non-integer, "fractional" dimension (for a general introduction, see [23]). NFDG is based on a generalization of the gravitational Gauss's law, replacing standard space integration over $\mathbb{R}^3$ with an appropriate Hausdorff measure over the space, which was related to Weyl's fractional integrals. As for MOND or the $\kappa$-model, the goal of NFDG is to describe galactic dynamics without using the controversial DM component. A quick review of NFDG is presented in reference [30]. NFDG was introduced heuristically by extending Gauss's law for gravitation to a lower-dimensional space-time $D + 1$, where $D \leq 3$ can be a non-integer space dimen-

sion. A scale length, $l_0$, is needed to ensure the dimensional correctness of all expressions for $D \neq 3$. Let us note that NFDG does not imply a change in the tri-dimensionality of space in galaxies, but rather the local Hausdorff dimension $D \neq 3$ is associated to matter distribution (bulge or disk). In this sense, there is an analogy with the $\kappa$-model, where the $\kappa$ ratios (Equation (1)) are assumed to be dependent on both the dimension of the matter distribution (bulge or disk) and also the compactness of matter (stars or gas).

In [29], Varieschi discusses the case of NGC 6503 in depth. For NFDG with dimension $D = 2$, the theoretical curve is slightly above the observational one and is remarkably flat (see Figure 6 of [29]). However, assuming that NGC 6503 behaves as a fractal medium, with a variable fractional dimension, NFDG can produce a curve with a perfect superimposition with the observational one. Referring now to Appendix A for this galaxy, we can see that both the MOND and the $\kappa$-model curves are slightly below the observational curve in the inner regions and slightly above in the outskirts. In the $\kappa$-model framework, the statement of variable fractional dimension could be re-interpreted as a variable thickness of the disk. In the case of NGC 6503, for instance, an increase in the thickness in the inner regions (thick disk) and, concomitantly, a decrease in the thickness in the outer regions (thin disk) could also lead to an improved profile, such as in NFDG theory. In [30], the same author applies his analysis to other rotationally supported galaxies, NGC 7814 and NGC 3741, for which very good NFDG fits are supplied. If we consider these galaxies, MOND and the $\kappa$-model provide a fairly good value for the terminal velocity. However, the same bias is perceptible for the inner velocities (the predicted curve is below the observational one). This bias is not present on the NFDG profiles which perfectly fit the corresponding observational curves, even with their humps and wiggles. However, this perfect fit results from the fact that the NFDG theory causes the fractional dimension to vary in "an appropriate manner" along a galactic radius in order to obtain a "good" profile. Nevertheless, the positive point of this procedure is that NFDG can be predictive for variable dimension. Once again, the $\kappa$-model can correct the mentioned bias by invoking a variable thickness. In the framework of this model, a volume effect, i.e., the influence of the mean volumetric mass density surrounding a given observer and estimated at a very large scale, is playing a similar role to that of the dimension in NFDG. Yet, Varieschi underlines that variable dimension $D$ should be interpreted as the dimension of the matter distribution of the galactic structure and definitely not at all as the local space dimension that an observer would measure at a specific galactic location. In any event, the link between a variable dimension in NFDG theory and a variable thickness in the $\kappa$-model could be more subtle, and should be reconsidered in greater depth. Furthermore, examining Relation (1), we can see that coefficients $\kappa$ for the bulge and the stellar and gaseous disks are different. For the bulge and the disk, the dimensions are admittedly different, but what about for the stellar versus gas components? All these questions deserve further examination.

It will be very interesting to apply the NFDG theory to a larger sample of galaxies for comparison with the $\kappa$-model; for instance, the totality of the galaxies of the SPARC database can be used. Particular attention must then be paid to the following cases: F563-V2, F568-1, F568-V1, F571-8, F579-V1, F583-1, NGC 2915, NGC 3992, NGC 5907, NGC 5985, NGC 6674, UGC 00128, UGC 00731, UGC 02259, UGC 6446, UGC 06667, UGC 07399, UGC 7490, and UGC 8286, for which both MOND and $\kappa$-model substantially differ from the observational profiles in the innermost regions, while, however, providing fairly good estimates on the outskirts of these galaxies (terminal velocities).

### 3.3. Refracted Gravity

Along with the NFDG model, another new classical gravity modified theory is the so-called Refracted Gravity (RG) [24,33,34]. RG can be reformulated as a scalar-tensor theory [34]. RG mimics DM with a gravitational permittivity (a kind of variable gravitational "constant" $G$), and that boosts the gravitational field in low-density environments. In RG, the link between volumetric mass density $\rho$ and gravitational permittivity $\epsilon$ is expressed by using relationship.

$$\epsilon(\rho) = \epsilon_0 + \frac{(1 - \epsilon_0)}{2} \left\{ tanh[ln(\frac{\rho}{\rho_c})^Q] + 1 \right\} \tag{6}$$

where $\epsilon_0$, $Q$, and $\rho_c$ are three free parameters. Formula (6) is an arbitrary monotonic function of the volumetric mass density with asymptotic limits $\epsilon(\rho) = 1$ for $\rho >> \rho_c$ and $\epsilon(\rho) = \epsilon_0$ for $\rho << \rho_c$. This formula is the equivalent in RG of Relation (2) in the $\kappa$-model. This permittivity also shares a very strong analogy with function $\mu$ in MOND [13], or still function $\kappa$ in the $\kappa$-model [16,26]. However, $\epsilon$ is supported by three universal parameters instead of just one, for instance, as in MOND ($a_0$). Thus, RG seems, at first glance, to be less economic than MOND, but its great feature of interest is that it is now a multiscale version of MOND. In this sense, the objective of RG is very similar to that proposed by the $\kappa$-model, but with an essential difference: the $\kappa$-model uses exclusively internal parameters (i.e., the mean volumetric mass densities of the bulge, stellar, and gaseous disk components) and no free external parameters. Then, by contrast, in RG, the three arbitrary parameters still need to be obtained through a long statistical analysis of the observational data [24]. RG has been applied to both flattened galaxies [24] and a small number of elliptical galaxies [33]. The results presented in [24] rely on setting the three free parameters for each individual galaxy. However, the authors show that the variations of these parameters from galaxy to galaxy can, in principle, be ascribed to statistical fluctuations. Then, the authors adopt an approximate procedure to estimate a single series of parameters that may properly describe the kinematics of the entire sample of galaxies. They eventually conclude that the gravitational permittivity is indeed a universal function. Unfortunately, a direct and yet fruitful comparison between RG and the $\kappa$-model is difficult because the galaxies under consideration are not issued from the same catalog. However, a close examination of the results displayed in [24] leads to the firm conclusion that the fits of the rotation profiles are of similar quality to those produced by MOND and the $\kappa$-model.

*3.4. Conformal Gravity*

Eventually, a comparison with conformal gravity can also be proven worthwhile. In the Conformal Gravity (CFT) [20] two universal parameters are introduced, setting apart the usual observational data, i.e., the luminosity and the $M/L$ ratio, the distance, and the inclination, common to any model. The first parameter ($\gamma*$) is related to the local geometry, while the second parameter ($\gamma_0$) describes the global geometry due to all the other galaxies in the Universe. These two parameters are statistically derived from the observational rotation curves of a chosen sample of 104 galaxies (this sample is limited to the galaxies whose mass density is fittable by a simple exponential thin disk). By comparison, we recall that in the $\kappa$-model the coefficients $\kappa$ are calculated from the mean mass density profiles attached to each galaxy. However, it is very difficult to decide which model is the best. Statistically, MOND, the $\kappa$-model and CFT provide equivalent results as for the proximity of the theoretical curves to the observational ones. We can examine the fit through the individual cases presented in [20]. For NGC 1003, NGC 3972, NGC 5585, and UGC 7089, MOND and $\kappa$-model fit is better than the CFT fit. For NGC 2903, UGC 5005,and UGC 5999 the fits are equivalent, For NGC 4100 the CFT fit is better than the MOND and $\kappa$-model fits, etc. Some cases are favorable to MOND or to the $\kappa$-model while in other cases the CFT is better. At the present time, this situation is very embarrassing because each author can validly support his own model against that of others through a judicious choice of the data. It is for that reason that the models have to be compared on a very large sample of galaxies such as the SPARC database, and not on a very small sample of a few galaxies.

**4. Conclusions**

This paper presents a discussion on the capacity of a number of MOND-type models [16,23,24] and a CFT-based model [20], which have been recently proposed to understand the dynamics of a large variety of flattened galaxies. Admittedly, these models do not provide very perfect fits (except maybe NFDG that possesses a flexible dimension asso-

ciated to mass distribution); nonetheless, they produce fairly predictive mean rotational curves. It is true that DM can indeed lead to better fits with two or three [44,45] parameters, but unfortunately, these parameters are freely adjusted to each galaxy. This implies that by starting from any Newtonian profile, even one strongly impaired by various biases, we can derive a "very good" fit for any given observational rotation profile[8]. MOND and the $\kappa$-model are at least falsifiable and upgradable, while DM is definitely not. For a physicist, the choice is apparent. With no confirmation by experimental methods, DM, unfortunately, has very limited scientific significance. Obviously, this conclusion will drastically change if, one day, we discover the signature of DM in the laboratory. We can always expect it over the next few years. Even though it is obvious that the $\kappa$-model is not a definitive solution, at least it shows that the baryonic mean mass density can play an unexpected role in the determination of the galactic rotational velocities and that both are strongly correlated. If this model is on the right track, then the rotational velocities alone can allow for us to directly determine the baryonic mean mass density (and not indirectly, using brightness measurements) and vice versa in a self-consistent manner. In this case, the delicate step, i.e., brightness $\rightarrow$ mean mass density, would be short-circuited. After an analysis of spiral galaxies and galaxy clusters, the work is far from finished. The $\kappa$-model still has to be applied to the elliptical galaxies, to the globular clusters, to the formation and stability of primordial galaxies, and eventually to CMB/cosmology. Let us also note the very captivating open debate concerning the wide binary stars ([54] versus [55]). The $\kappa$-model obviously predicts a very weak $\kappa$ effect in the immediate vicinity of the Sun, i.e., the motion of the wide binaries is predicted to be quasi-Keplerian in this region. Much work remains to be performed. It would be interesting to concomitantly conduct the same studies on the same collection of galaxies with other models such as the Newtonian Fractional-Dimension Gravity [29–32], the Refracted Gravity [33,34], and also the CFT model [20].

**Funding:** This research received no external funding.

**Data Availability Statement:** The author confirms that the data supporting the findings of this study are available within the article and the reference list.

**Acknowledgments:** The author sincerely thanks the three referees whose suggestions and corrections improved this paper.

**Conflicts of Interest:** The author declares no conflict of interest.

## Appendix A. Section Rotation Curve Fits Results

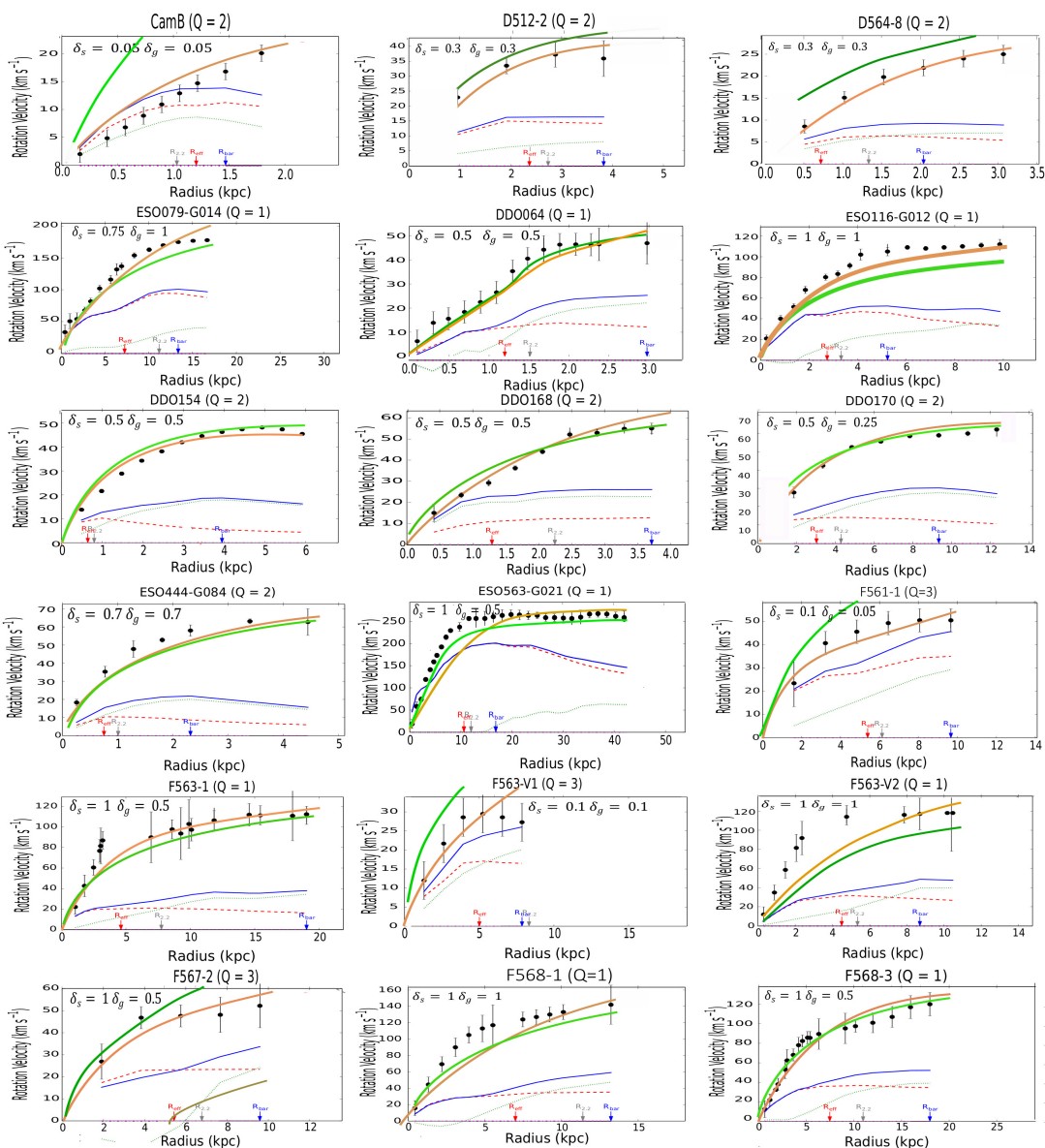

**Figure A1.** Rotation curves of the SPARC galaxies. The green line is predicted by MOND, the amber line is predicted by the $\kappa$-model, the red dashed line represents the stars, the green dotted line represents the gas, the blue line represents the sum of all baryonic components (stars + gas). The observed velocities are shown as a series of filled black circles.

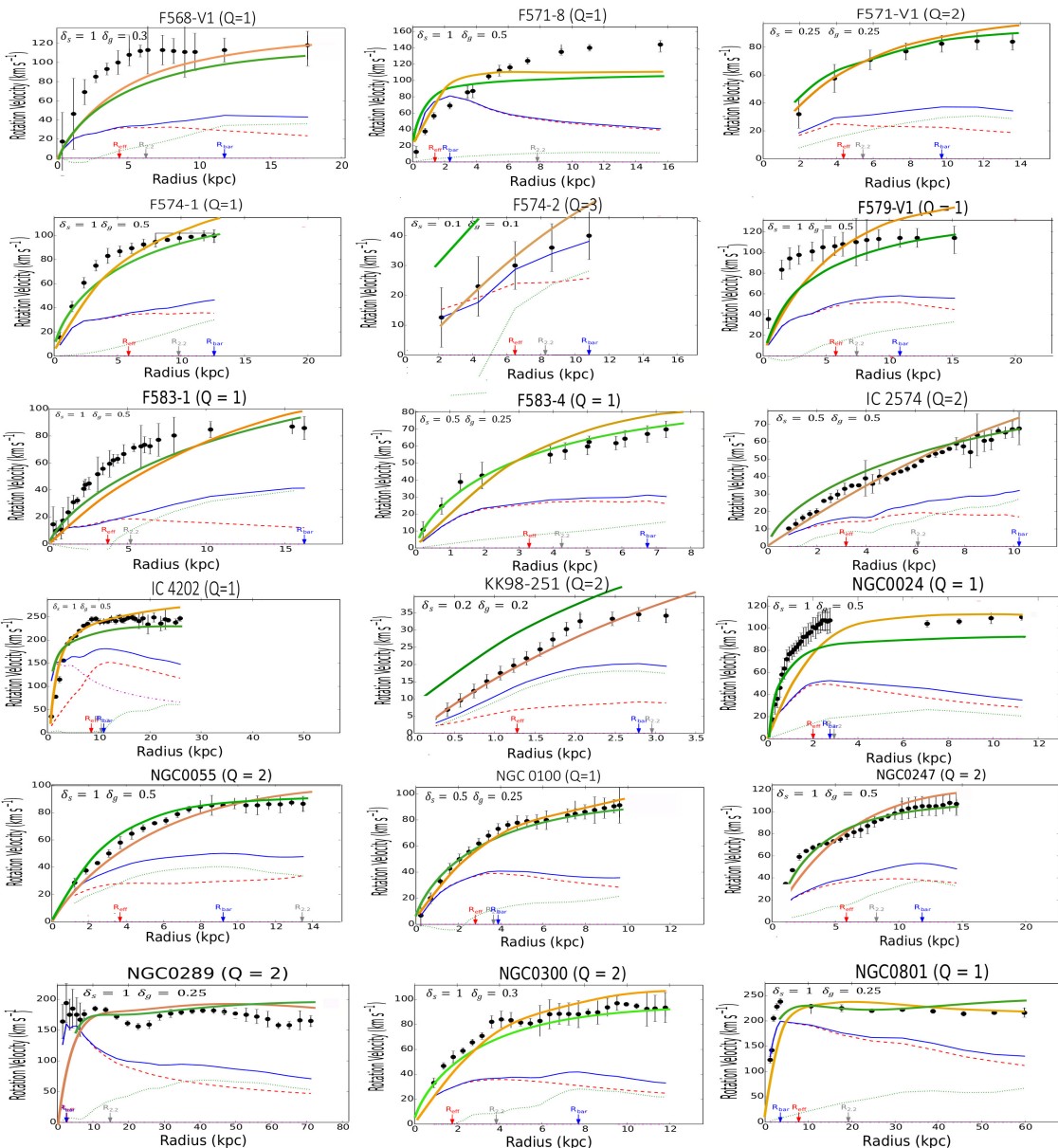

**Figure A2.** Continued rotation profiles.

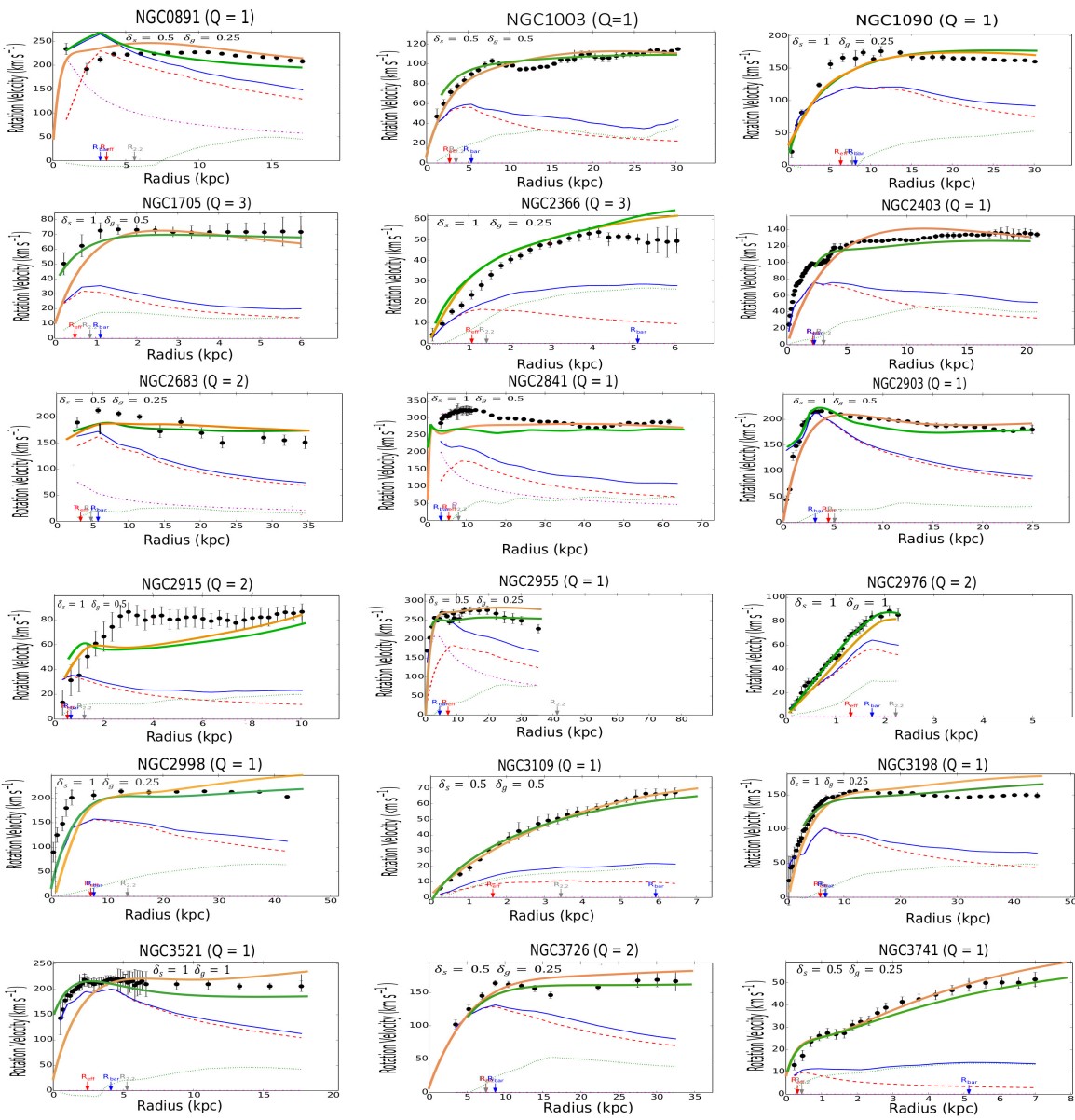

**Figure A3.** Continued rotation profiles.

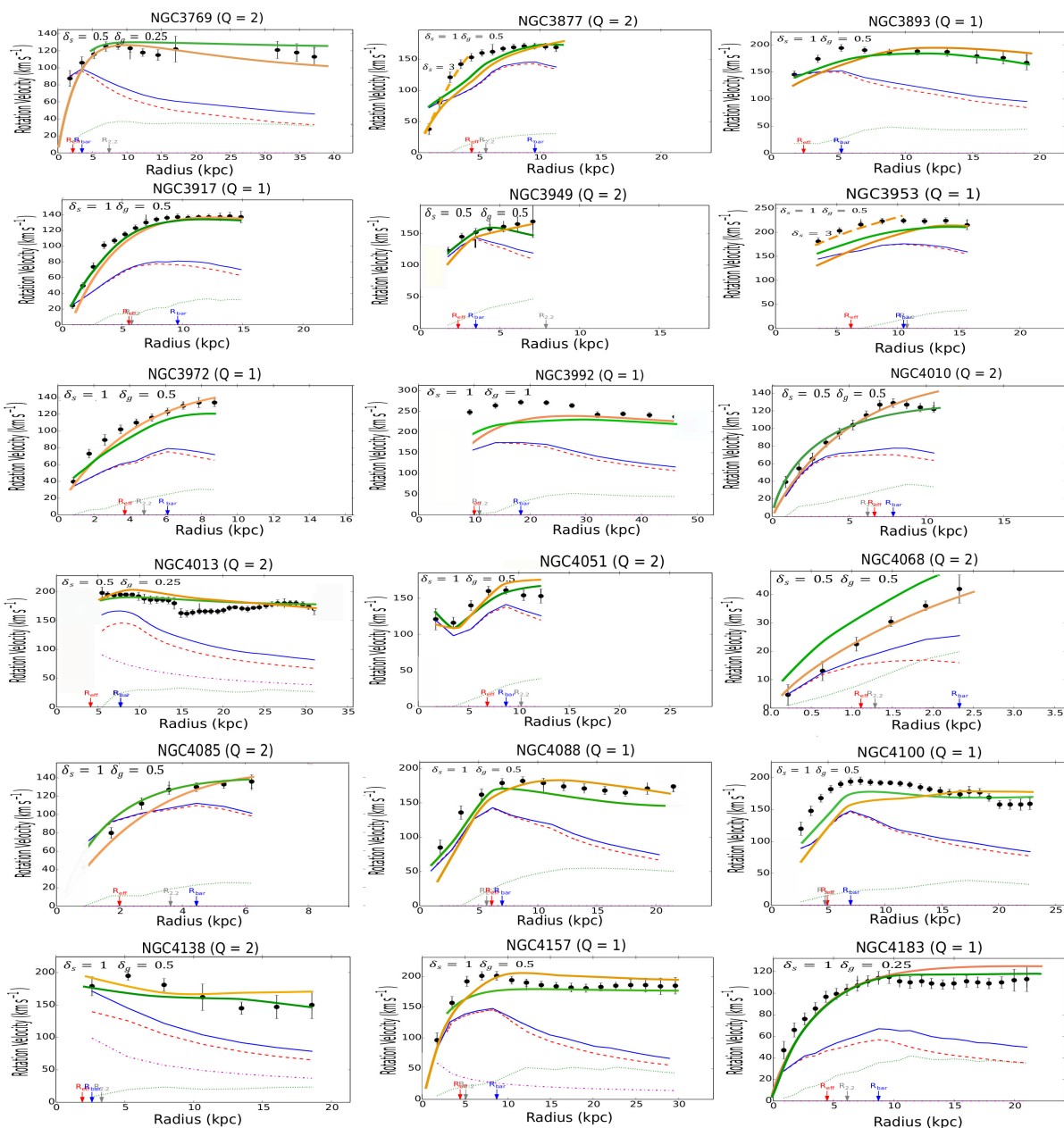

**Figure A4.** Continued rotation profiles.

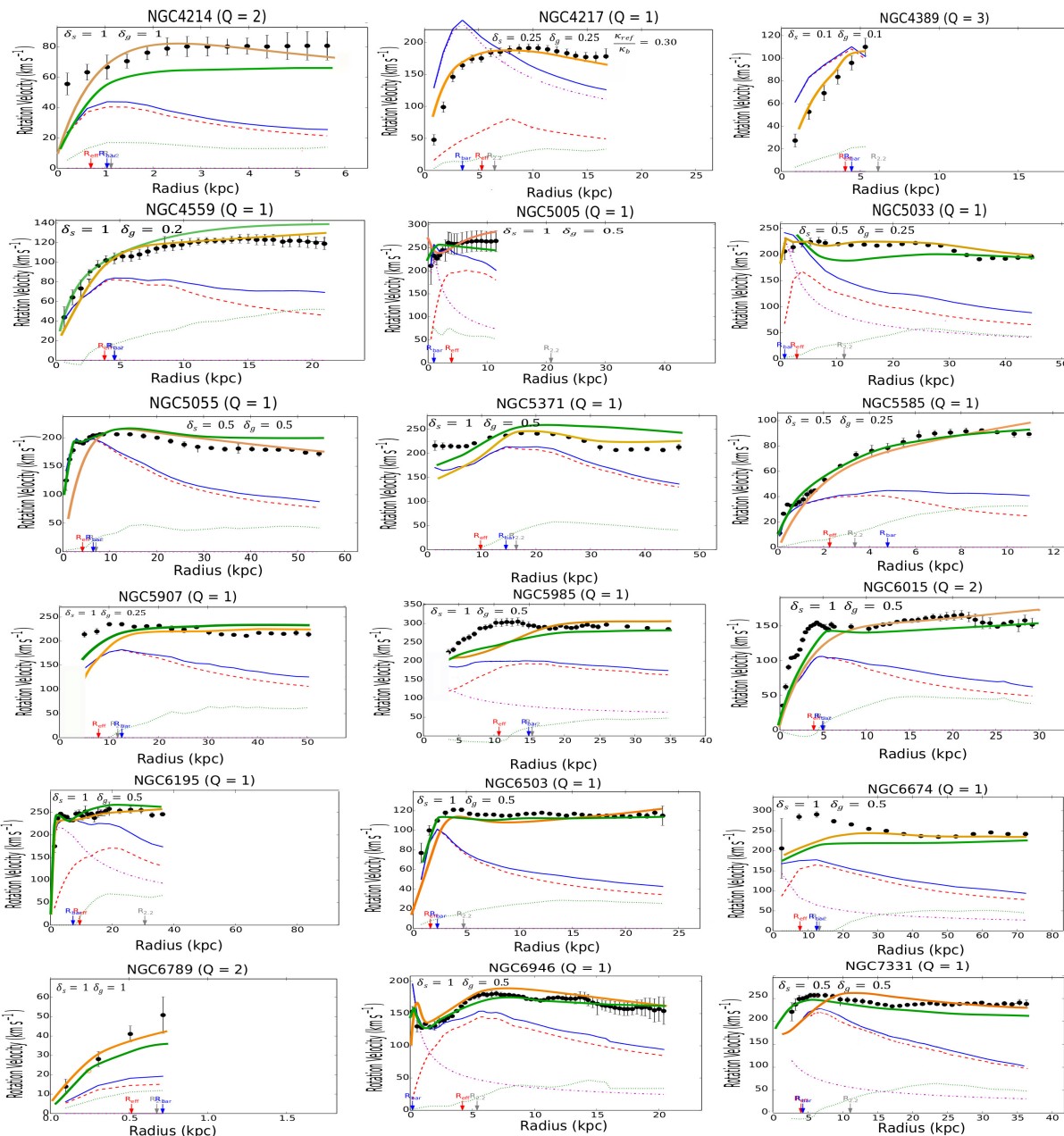

**Figure A5.** Continued rotation profiles.

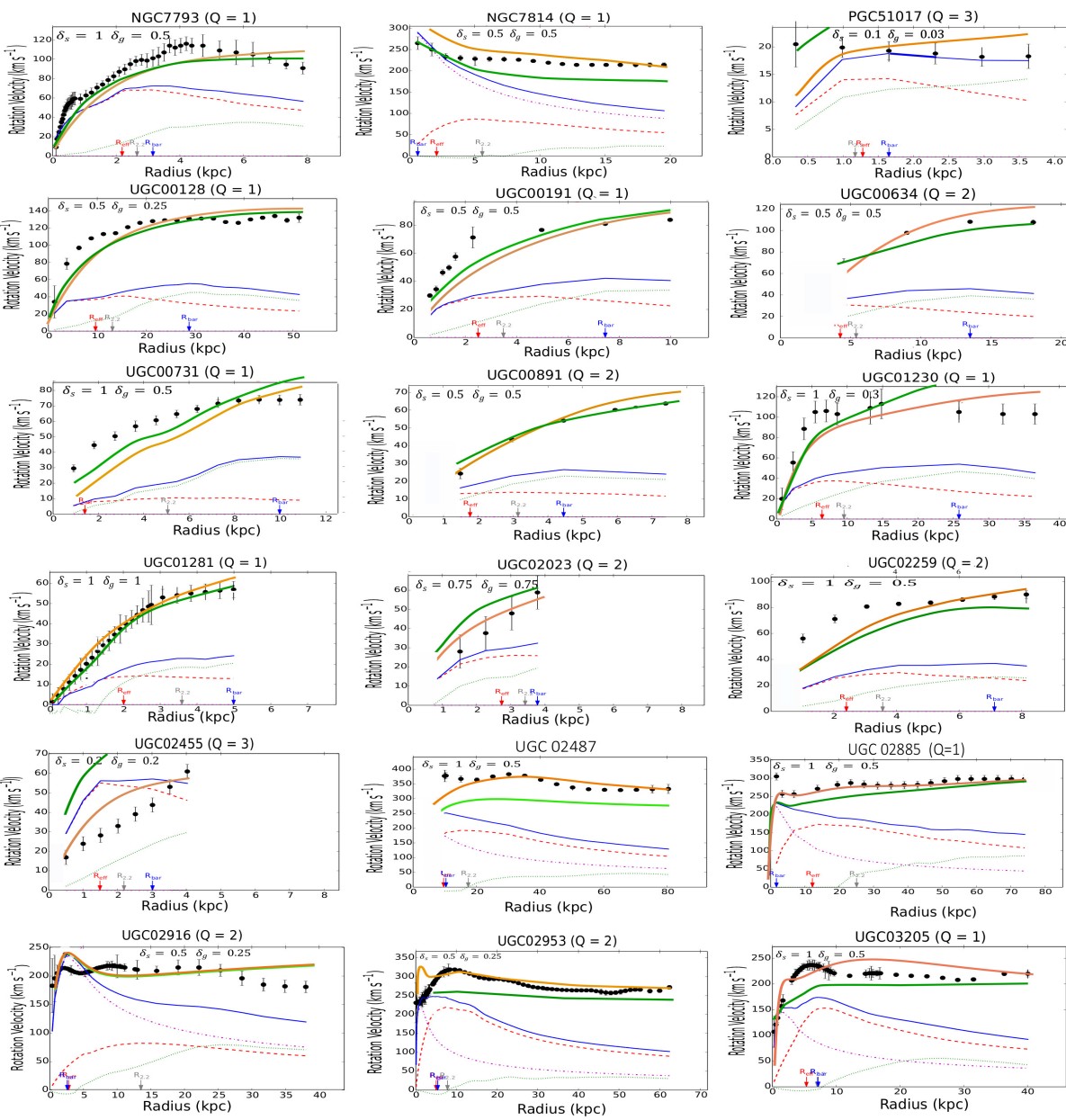

**Figure A6.** Continued rotation profiles.

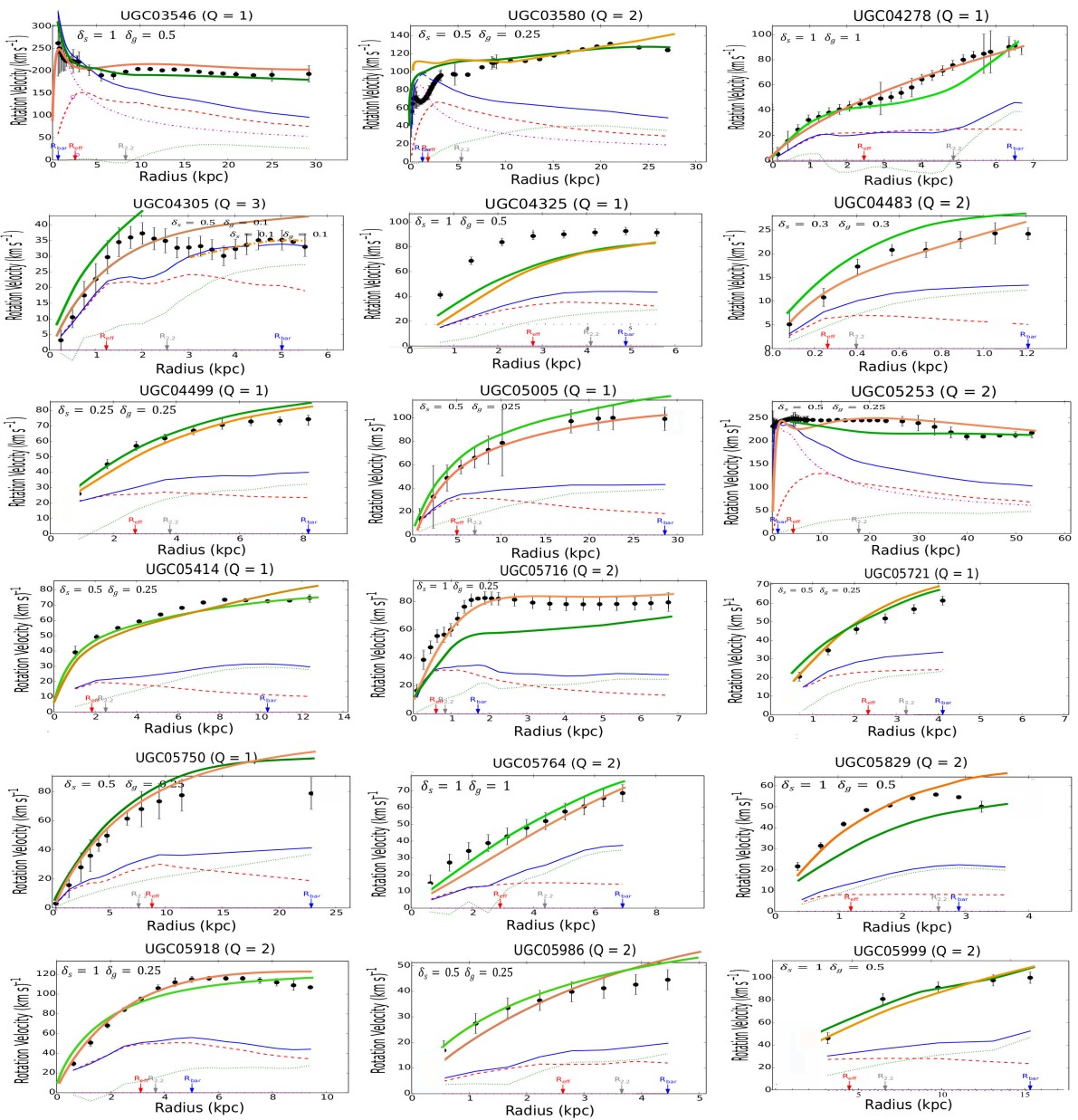

**Figure A7.** Continued rotation profiles.

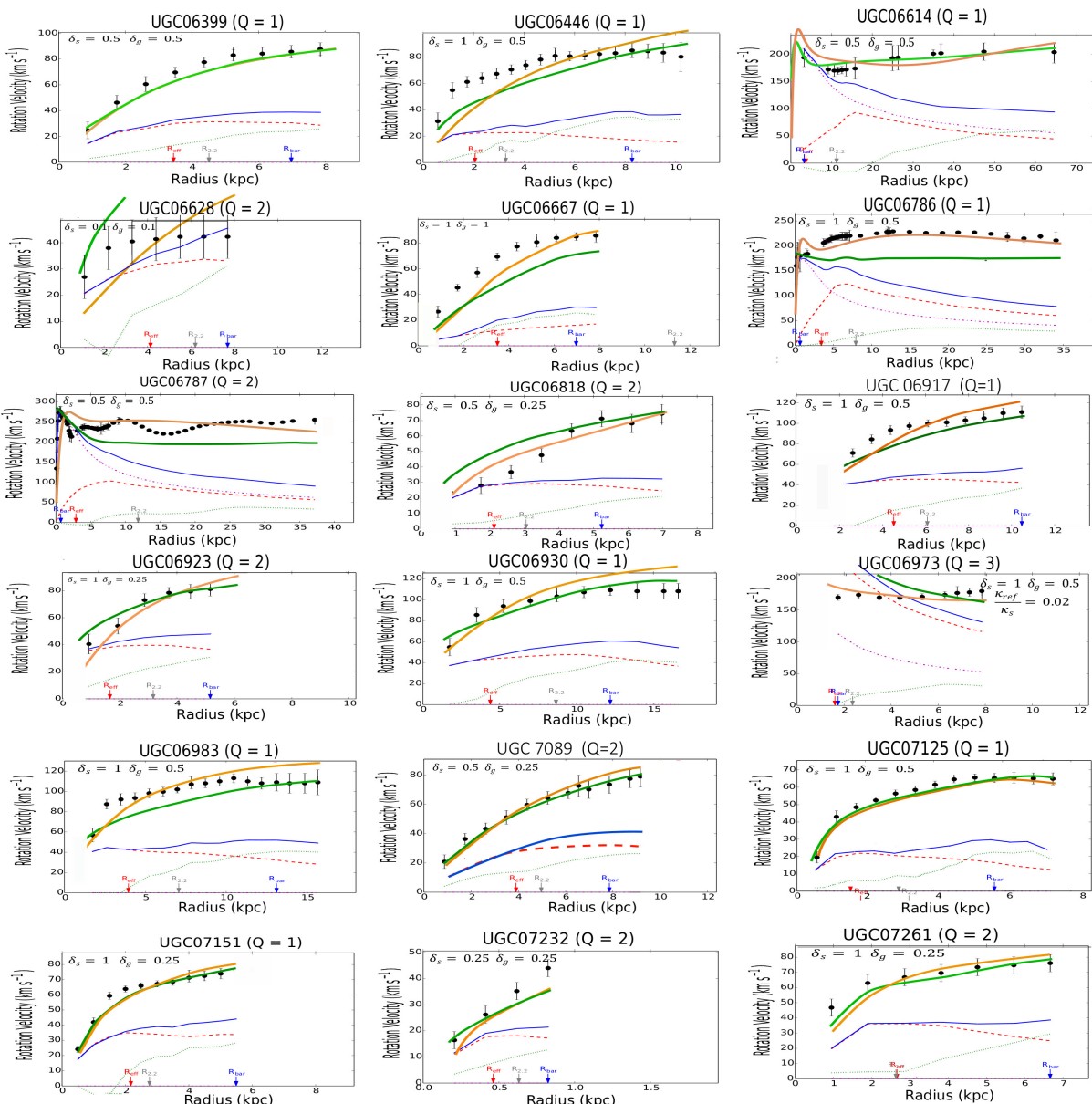

**Figure A8.** Continued rotation profiles.

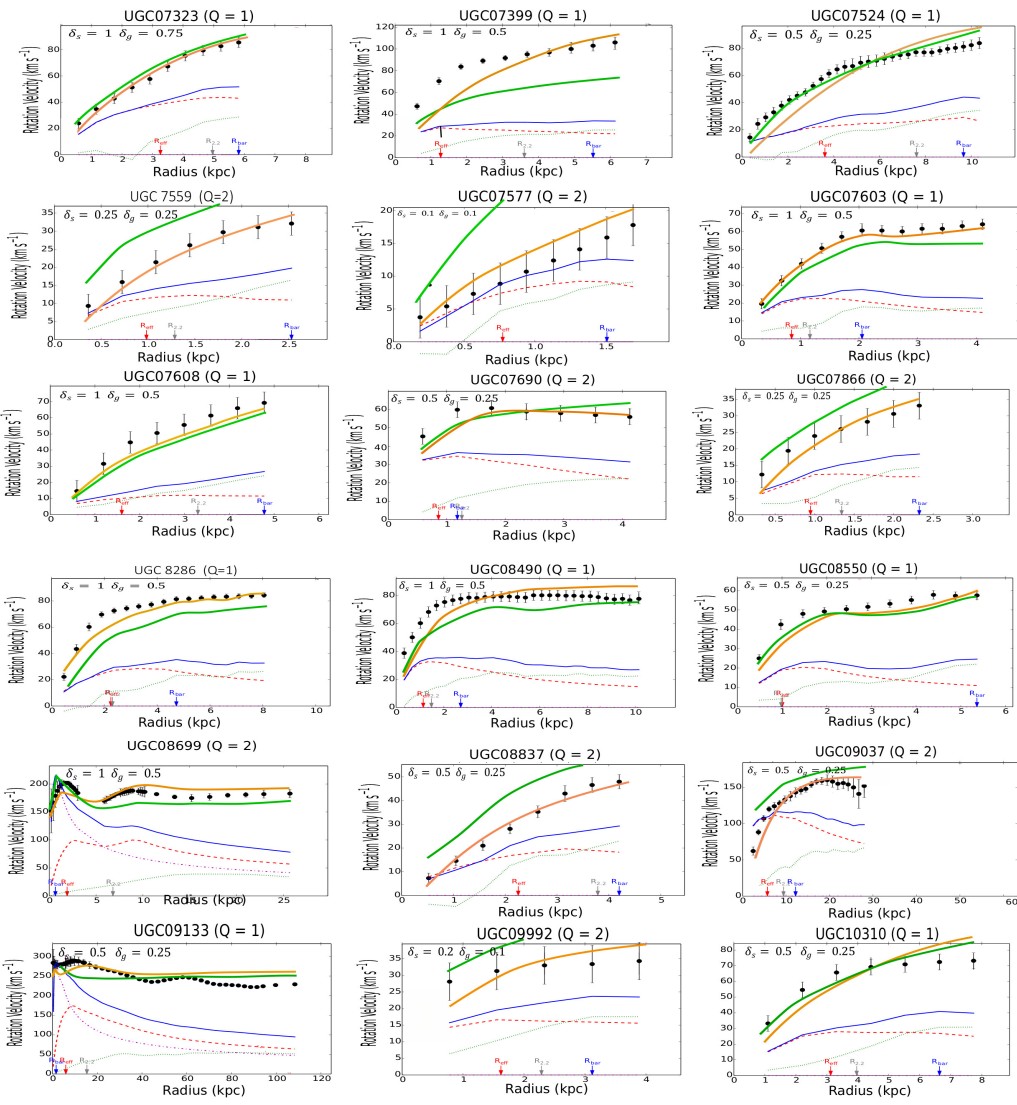

**Figure A9.** Continued rotation profiles.

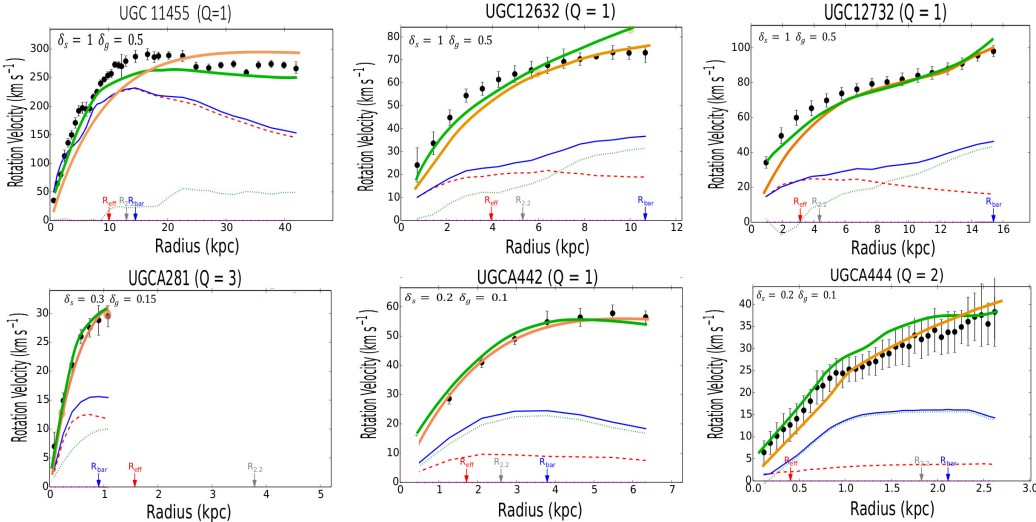

**Figure A10.** Continued rotation profiles.

## Notes

[1]　In the context of MOND, a multiscale approach adapts parameter $a_0$ to the size of the system studied. However, in this case, parameter $a_0$ is no longer universal

[2]　Performance of the $\kappa$-model with lensing is presented in reference [16].

[3]　The Weyl CFT is built by replacing the Einstein–Hilbert Lagrangian density, proportional to the Ricci curvature scalar, by a quadratic contraction of the conformal Weyl tensor.

[4]　A superlens is a flat, lightweight option that can replace bulky traditional lenses and other components in optical systems. It is a lens that moves beyond the diffraction limit.

[5]　A list of DM methods with two or three ad hoc parameters is presented in references [43,44].

[6]　The MOND profiles are obtained with formula

$$v^2_{MOND} = v^2_{bew}\left[\frac{1}{2} + \frac{1}{2}\left[1 + 4\left(\frac{a_0}{g_{new}}\right)^2\right]^{\frac{1}{2}}\right]^{\frac{1}{2}} \tag{5}$$

where $v_{new}$ and $g_{new}$ are, respectively, Newtonian velocity and acceleration.

[7]　We can remark that whereas some authors affirm that the MOND fits are fairly good [47,49], on the other hand, other authors, who rather seem to defend the DM paradigm, conclude that the MOND fits are bad in a large percentage of analyzed individual cases [50,51]. It is true that MOND offers fairly acceptable fits in a large number of cases and is worse in other cases. This situation can easily be explained if we admit that the mass-to-light ratios, the inclinations, and the distances are poorly known. We can then postulate that MOND systematically provides "perfect" fits and can then predict the inclinations and distances. In contrast, DM fits are apparently better but, ironically enough, even if we choose a bad inclination or an erroneous distance.

[8]　Once again, in the framework of the $\kappa$-model starting from the Newtonian curves, we generate a predictive profile for the observational one in an univocal manner. The baryonic mean mass density alone is the conductor of the situation.

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
