# Peer review of "The κ-Model under the Test of the SPARC Database"

_universe, doi:10.3390/universe10030151_

Round 1

Reviewer 1 Report

Comments and Suggestions for Authors

The manuscript compares the kappa-model to MOND and other models for to the DM model in fitting rotation curves of SPARC galaxies. It has rather detailed introduction in the problem of the different models which I think would be interesting for people outside the narrow field. I recommend the paper for recommendation after major revision.

The over-all note is that the paper is mostly a comparison paper between different models. I suggest that all these models get a proper introduction in the text with the formulas for the velocity and the discussed parameters written the way Eq. 1 is written.  I suggest adding a subsection to section 2, with the equations of DM, MOND, kappa-model and maybe of NFDG and RG, so that the reader can easily compare the models and their parameters.

Also, the abstract talked about the universal parameter, but the text almost doesn't treat this issue. There's one paragraph about it (96-102) so I suggest elaborating on it where relevant when presenting the models and maybe in the conclusion. From what I saw from the plots, it seems like the kappa-model and MOND give more or less similar predictions in most cases, the paper outlines some of the reasons for the deviations, maybe that's worth being summarized in the conclusion as well. 

The plots in Figure 1 seem stretched and not in the right resolution, so if possible please fix it. 

Finally, there is no mentioning of how the plots have been obtained - like numerical methods or assumptions made, so adding a sentence about that would be good for the paper. 

Below are my notes from reading the full text:

line 80: Is missing mass problem? -> Is it a missing mass problem

88: that DM -> than DM

99: trial particle -> test particle

121: please explain the multiscale approach

145: with a fairly efficiency -> fairly efficiently

178: with respect to QG, the most recent review is Addazi et al. Quantum gravity phenomenology at the dawn of the multi-messenger era—A review"

192: Maybe it would be interesting to include an estimation about how much the DM parameters vary across different galaxies. (this is just a suggestion)

196: . ->:

199: seeing -> view, understanding

The paragraph starting with line 200: it is a suggestion, but I think the use of "his" can be replaced with "theirs" to make everyone happy.

212: metalens?

213: without to be -> without being

230: is there a reference to the relativistic kappa-model. If it's simple and straight forward why not use the relativistic version instead? How good is the Newtonian approximation in this case. 

I think footnote 2 belongs in the text

261: do indexes "1,2" run on all the mentioned indexes, or something else.

264: "In relation to" -> "In relation to the indexes" 

285-286: could you elaborate on this, for people who are not specialist

292: should to -> should

296: notified -> noticed

299: reference

311: free-parameter -> parameter-free

333: 1) BAO should be written out fully since it's not used anywhere else in the text and 2) this explanation of BAO is a bit minimalistic and odd, since while BAO are an effect from the recombination epoch, BAO are observed in galaxies distributions. I suggest explaining it properly. 

340 Maybe a reference for this? 

341: surprisingly close? Compared to what, since it's about twice as much. 

360: "a very similar role" - similar to the logarithm - it is not written very clearly/

369: an example of object with this high surface density

379: apart? Maybe "different"? (the sentence is unclear)

380: unique exponential disk? (the sentence is unclear)

435:  I think here or if there is a section on DM, it would be useful to use the actual names of the models with different parameters.  

506: definition of EFE

554: more deep -> deeper, maybe -> maybe

556: concern -> interest?

557: Is  these galaxies referring to the list of galaxis on line 511-513? Written like this is unclear. 

643: put the abbreviation in brackets the first time it appears (RG). Refracted gravity is mentioned on line 187 so try to make the two places it's discussed coherent and avoid repetitions. 

671: please explain the connection with cosmology and please use the names of the parameters for clarity.

692: "a thinking"? "A discussion", maybe is better.

701: something is missing near the footnote because it feels like "procedure" should be part of new sentence.

I feel like The Discussion section can be divided into subsections - 1) comparisong bettween kappa-model and MOND and 2) comparison with other theories. It would improve the readability. Right now there's no formatting of the text and it's not easy to read. The use of subsections, lists or bold text would make the manuscript more attractive. 

Comments on the Quality of English Language

Please pass the text trough some kind of grammar check, because I listed a lot of typos in my notes, but there surely are more. 

Author Response

Dear  reviewer,

Following your  comments,  the point-by-point corrections  appear in red. I have passed the text through a grammar checker.

I would like to thank you  for carefully reading my manuscript and for your comments.

Reviewer 2 Report

Comments and Suggestions for Authors

Dear Author,

the underlying manuscript adds important aspects to the ongoing discussion about the origin of the apparent excess gravity, which is being addressed to dark matter in the standard model of cosmology, but can also be described by variation of the law of gravity from Newtonian/Einsteinian gravity.

The scientific content is, to my impression, an important discussion of alternative models to DM, namels MOND, the kappa model, Refracted Gravity, Newtonian Fractional-Dimension Gravity, and Conformal Gravity. Of these, however, only MOND and the kappa model are extensively examined, which might be stated more clearly in the abstract.

Gravitational lensing is mentioned briefly a few times, but it would be interesting to know, how the kappa model performs with lensing. This, however, may be addressed in an upcoming study.

There remain a few other issues, mainly of style rather than scientific. There are repeated typographical errors like extra spaces, differential symbols (dx, dy) and units (km/s) in italics rather than upright as well as a few typos.

In equation (2), does "Ln" mean the natural logarighm? Please write "ln" (in lower case and in upright font shape rather than italics).

Comments on the Quality of English Language

Please avoid colloquial or slang wordings like "all this stuff" (line 629) in a scientific article.

Author Response

Dear  reviewer,

Following your  comments,  the point-by-point corrections  appear in blue.

I would like to thank you  for carefully reading my manuscript and for your comments.

Reviewer 3 Report

Comments and Suggestions for Authors

In this paper the author compares three recent alternative gravity models: his own kappa-model, NFDG (Newtonian Fractional-Dimension Gravity), and RG (Refracted Gravity). This comparison is carried out with respect to the galaxy rotation curves obtained from the SPARC database. All three models aim to describe these galactic data without using any dark matter component and they all have connections with Milgrom’s MOND theory. In particular, the kappa-model seems to be effective in fitting most of these rotations curves, at least at the same level of the MOND model.

Possible suggestions/corrections regarding this manuscript:

1.       The general description of the kappa-model in section 1 and the fundamental equations (1) and (2) on page 10 should be improved, by adding more details about the origin of all the kappa coefficients. These were introduced by the author in previous publications, such as [31]-[32], but more details could be added also in the current paper.

2.       In addition to the already cited NFDG papers [22],[26-28] in the manuscript references, another NFDG paper of interest might be: Mon.Not.Roy.Astron.Soc. 503 (2021) 2, 1915-1931 e-Print: 2011.04911 [gr-qc], where two other galaxies from SPARC database (NGC 7814 and 3741) were considered. These could be added to the discussion/comparison on page 23 between k-model and NFDG results.

3.       The manuscript should be double-checked to improve the English language and correct many typos and language issues.

This manuscript will probably be of interest to the readers of Universe and I recommend publication after addressing all points mentioned above.

Comments on the Quality of English Language

    The manuscript should be double-checked to improve the English language and correct many typos and language issues.

Author Response

Dear  reviewer,

Following your  comments,  the point-by-point corrections  appear in orange. The  text has been double-checked.

I would like to thank you  for carefully reading my manuscript and for your  comments.

Round 2

Reviewer 1 Report

Comments and Suggestions for Authors

The author has taken into account most of my comments in the revised version and the English has been improved, so I think the manuscript can be published in the current form.

I noticed just a few cosmetic things and this is why I picked "Minor revision":

266 - The table is nice, but maybe put as a header row "model"/"main features" 

405 - "patent fact" sounds strange to me

542 Figure A -> Figure A in the Appendix

552 Fig. 2 -> Figure 2

Author Response

Dear reviewer,

I have taken careful note of your comments.